



**Quantifying SO₂ oxidation pathways to atmospheric sulfate by using**
**stable sulfur and oxygen isotopes: laboratory simulation and field**
**observation**
Ziyan Guo[a], Keding Lu[a]*, Pengxiang Qiu[b], Mingyi Xu[b], Zhaobing Guo[b*]
[a] State Key Joint Laboratory of Environmental Simulation and Pollution Control,
State Environmental Protection Key Laboratory of Atmospheric Ozone Pollution
Control, College of Environmental Sciences and Engineering, Peking University,
Beijing, China.
[b] Jiangsu Key Laboratory of Atmospheric Environment Monitoring and Pollution
Control (AEMPC), Collaborative Innovation Center of Atmospheric Environment and
Equipment Technology (CIC-AEET), School of Environmental Science and
Engineering, Nanjing University of Information Science and Technology, Nanjing
210044, China
*Correspondence to: k.lu@pku.edu.cn (Keding Lu), guocumt@nuist.edu.cn
(Zhaobing Guo)



**Abstract.** The formation of secondary sulfate in the atmosphere remains controversial, and it is urgent
to seek for a new method to quantify different sulfate formation pathways. Thus, $SO_2$ and $PM_{2.5}$
samples were collected from 4 to 22 Dec. 2019 in Nanjing. Sulfur and oxygen isotope compositions
were synchronously measured to study the contribution of $SO_2$ homogeneous and heterogeneous
oxidation to sulfate. Meanwhile, the correlation of $\delta^{18}O$ values between $H_2O$ and sulfate from $SO_2$
oxidation by $H_2O_2$ and $Fe^{3+}/O_2$ were investigated in the lab. Based on isotope mass equilibrium
equations, the ratios of different $SO_2$ oxidation pathways were calculated. The results showed that
secondary sulfate constituted higher than 80 % of total sulfate in $PM_{2.5}$ during the sampling period.
Laboratory simulation experiments indicated that $\delta^{18}O$ of sulfate was linearly dependent on $\delta^{18}O$ of
water, and the slopes of linear curves for $SO_2$ oxidation by $H_2O_2$ and $Fe^{3+}/O_2$ were 0.43 and 0.65,
respectively. The secondary sulfate in $PM_{2.5}$ was mainly ascribed to $SO_2$ homogeneous oxidation by
OH radicals and heterogeneous oxidation by $H_2O_2$ and $Fe^{3+}/O_2$. $SO_2$ heterogeneous oxidation was
generally dominant during sulfate formation, and the contribution of $SO_2$ heterogeneous oxidation was
about 52 %. Especially, $SO_2$ oxidation by $H_2O_2$ predominated in $SO_2$ heterogeneous oxidation reactions
with an average ratio around 55 %. This study provided an insight into precisely evaluating sulfate
formation pathways by combining stable sulfur and oxygen isotopes.








**1 Introduction**

Sulfate is one of the prevalent components of $PM_{2.5}$ (Brüggemann et al., 2021; Huang et al., 2014; Yang et al., 2023). Sulfate makes up approximately 25% of $PM_{2.5}$ mass in Shanghai, 23% in Guangzhou and 10-33% in Beijing (Xue et al., 2016). The rapid sulfate formation is a crucial factor determining the explosive growth of fine particles and the frequent occurrence of severe haze events in China (Lin et al., 2022; Liu et al., 2020; Meng et al., 2023; Wang et al., 2021). Sulfate plays an important role in tropospheric and lower stratospheric chemical and physical processes, which significantly affects global climate change by scattering solar radiation and acting as cloud condensation nuclei (CCN) (Gao et al., 2022; Ramanathan et al., 2001). Meanwhile, sulfate exerts a significant influence on air quality and public health (Abbatt et al., 2006).

In the past decades, numerous attempts have been made to evaluate $SO_2$ oxidation pathways involving in homogeneous and heterogeneous reactions. Traditionally, sulfate formation mechanisms mainly include homogeneous oxidation of $SO_2$ by OH radicals and heterogeneous oxidation by $H_2O_2$, $O_3$ and $O_2$ catalyzed by transition metal ions (TMIs) in cloud/fog water droplets. The relative importance of different sulfate formation pathways is strongly dependent on oxidant concentrations, occurrence of fog/cloud events and pH of aqueous phase (Seinfeld et al., 2016; Kuang et al., 2022; Oh et al., 2023). Generally, $SO_2$ homogeneous oxidation by OH and heterogeneous oxidation by $H_2O_2$ are considered the most important pathways for sulfate production on the global scale (Seinfeld et al., 2006). The photochemical reactivity during the winter in Beijing has been found to be relatively high, which favors the formation of reactive species such as OH radicals and $H_2O_2$, thereby facilitating $SO_2$ oxidation (Zhang et al., 2020). Xue et al. (2014) suggested that $SO_2$ oxidation by $O_3$ and $H_2O_2$ in aqueous phase contributed to the majority of total sulfate production. Liu et al. (2020) proposed that S(IV) oxidation by $H_2O_2$ in aerosol water could be an important pathway considering the ionic strength effect. He et al. (2018) found that the contribution of $SO_2$ oxidation by $H_2O_2$ could reach 88 % during Beijing haze period. Ye et al. (2018) observed that $SO_2$ oxidation rate by $H_2O_2$ was 2-5 times faster than the summed rate of the other three oxidation pathways. As a result, actual contribution of $SO_2$ oxidation by $H_2O_2$ during the winter might be underestimated in previous studies.

In addition, the presence of $NO_2$ was obviously favorable for $SO_2$ oxidation under the conditions of high RH and $NH_3$. $NH_3$ can promote the hydrolysis of $NO_2$ dimers to HONO and result in more sulfate



formation on particle surface in humid conditions. However, this conclusion was doubted by Liu et al.
(2017) who believed that the reaction on actual fine particles with pH at 4.2 was too slow to account
for sulfate formation. Li et al. (2020) deemed that $SO_2$ oxidation by $NO_2$ might not be a major
oxidation pathway in China. Furthermore, GEOS-Chem modeling study suggested that $NO_2$ oxidation
contributed less than 2% of total sulfate production. It is found that TMI pathway was very important in
highly polluted regions, and the contribution of metal-catalyzed $SO_2$ oxidation to sulfate was as high as
49±10% in haze. Wang et al. (2021) also argued that $SO_2$ oxidation by TMI on aerosol surface could be
the dominant sulfate formation pathway. They found that manganese-catalyzed oxidation of $SO_2$
contributed 69.2±5.0% in sulfate production. Overall, the mechanisms for sulfate rapid growth remain
unclear and controversial. Therefore, sulfate formation pathways via $SO_2$ oxidation need to be further
explored, and it is urgent to develop a new method to quantify different sulfate formation processes.
Generally, sulfur isotopes allow for investigating $SO_2$ oxidation processes in the atmosphere because
of distinctive isotope fractionation associated with different oxidation reactions (Harris et al., 2013).
Harris et al. (2012) presented sulfur isotope fractionation factors of $SO_2$ oxidation by OH, $O_3/H_2O_2$ and
iron catalysis were 1.0087, 1.0167 and 0.9905, respectively. Besides, the observed sulfur isotope
fractionation of $SO_2$ oxidation by $H_2O_2$ and $O_3$ appeared to be no significant difference. Therefore, the
results were particularly useful to determine the importance of transition metal-catalyzed oxidation
pathway compared to other oxidation pathways. However, other main oxidation pathways of $SO_2$ could
not be distinguished only based on stable sulfur isotope determination.
Oxygen isotope ratio ($\delta^{18}O$) can be used to deduce sulfate formation processes due to those different
SO2 oxidation pathways affect oxygen isotope of product sulfate differently. Especially,
mass-independent fractionation signals (nonzero $\Delta^{17}O$, where $\Delta^{17}O=\delta^{18}O-0.52\times\delta^{17}O$) of oxygen
isotopes in sulfate are usually adopted to investigate the contributions of different $SO_2$ oxidation
pathways. This method can identify the contribution of $SO_2+O_3$ pathway when high $\Delta^{17}O$ (>3‰) is
measured in sulfate. However, there is presence of obvious uncertainty when interpreting the sulfate
with low $\Delta^{17}O$ value (<1‰). Unfortunately, most sulfate samples in the atmosphere show $\Delta^{17}O<1‰$,
suggesting a limited contribution of $SO_2+O_3$ pathway during sulfate formation. It is noteworthy that the
contribution of $SO_2+H_2O_2$ pathway and TMI pathway is unclear if solely using $\Delta^{17}O$ (Li et al., 2020).
Holt et al. (1982) found oxygen isotope was a valuable and complementary method to determine



probable mechanisms of $SO_2$ oxidation to sulfate in the atmosphere. This provides us an insight into
precisely evaluating sulfate formation pathways by combining oxygen and sulfur isotopes.
In this contribution, $PM_{2.5}$ samples were collected from 4 to 22 Dec. 2019 in Nanjing region. Sulfur
and oxygen isotope compositions in sulfate were measured to study the contribution of $SO_2$
homogeneous and heterogeneous oxidation during sulfate formation. In addition, the linear
relationships of $\delta^{18}O$ values between $H_2O$ and sulfate from $SO_2$ oxidation by $H_2O_2$ and $Fe^{3+}/O_2$ were
synchronously investigated in the lab. Based on sulfur and oxygen isotope mass equilibrium equations,
the ratios of different $SO_2$ oxidation pathways during the sampling period were calculated. The study
aims to seek for a novel method to quantify different $SO_2$ oxidation processes with sulfur and oxygen
isotopes.
**2 Materials and methods**
2.1 Sampling location
$PM_{2.5}$ and $SO_2$ in the atmosphere were sampled from 4 to 22 Dec. 2019 in Nanjing, China. The
sampling site was located at the roof of the library in Nanjing University of Information Science &
Technology (NUIST, 32.1 °N, 118.5 °E), which is depicted in Fig. 1. The sampling location is at the
side of Ningliu Road and closely next to Nanjing chemical industry park. There is presence of some
large-scale chemical enterprises such as Nanjing steel plant, Nanjing thermal power plants and Nanjing
petrochemical company, which inevitably release lots of $SO_2$ and iron metal into the atmosphere.
2.2 $PM_{2.5}$ and $SO_2$ Samples collection
$PM_{2.5}$ and $SO_2$ were sampled using a modified JCH-1000 sampler (Juchuang Co., Qingdao) with a
flow rate of 1.05 $m^3\,min^{-1}$ from 8 am to 8 pm from 4 to 22 Dec. 2019. $PM_{2.5}$ and $SO_2$ were collected
with quartz filter (203×254 mm, Munktell, Sweden) and glass fiber filter (203×254 mm, Tisch
Environment INC, USA), respectively. The filters were incinerated in a muffle furnace at 450 ℃ for 2h
and then preserved in the desiccators at room temperature. The glass fiber filters were firstly soaked in
2% $K_2CO_3$ and 2% glycerol solution for 2h and dried in DGG-9070A electric oven. $SO_2$ can be
changed into sulfite immediately during the sampling.
2.3 Extractions of water-soluble sulfate
$PM_{2.5}$ sample filters were shredded and soaked in 400 mL of Milli-Q (18 MΩ) water for extractions



of water-soluble sulfate. Filters were then isolated from solutions by centrifugation and water-soluble
sulfate was precipitated as $BaSO_4$ by adding 1 mol $L^{-1}$ $BaCl_2$. After the filtration with 0.22 μm acetate
membrane, $BaSO_4$ precipitate was rinsed with Milli-Q water to remove $Cl^-$. Finally, $BaSO_4$ powers
were calcined at 800 ℃ for 2h to obtain high purity $BaSO_4$. In addition, a small amount of $H_2O_2$
solution was added to oxidize sulfite to sulfate.
2.4 Laboratory simulation of $SO_2$ oxidation by $H_2O_2$ and $Fe^{3+}/O_2$
For $SO_2$ oxidation by $H_2O_2$, 30 mL $min^{-1}$ Ar was firstly introduced into three kinds of different water
about 30 min to drive out air. Sulfate was produced by adding 10 mL $H_2O_2$ dilute solution (0.1 mL 30%
$H_2O_2$ in 50 mL water) to $SO_2$ in the reaction chamber at 10 ℃. $H_2O_2$ solution was agitated vigorously
for 1min before admission of air. For $Fe^{3+}$ catalyzed oxidation of $SO_2$, 2 mL $min^{-1}$ $SO_2$ and 2 mL $min^{-1}$
$O_2$ were simultaneously put into $Fe^{3+}$ dilute solution at 10 ℃. Then, 10 mL 1 mL $min^{-1}$ $BaCl_2$ was
added to prepare $BaSO_4$. Oxygen isotope compositions of product sulfate and three kinds of water were
measured to study their linear relationships.
2.5 Sulfur and oxygen isotope determination
Sulfur isotope compositions in sulfate were analyzed using Elemental analyzer (EA, Flash 2000,
Thermo) and isotope mass spectrometer (IRMS, Delta V Plus, Finningan). High-purity $BaSO_4$ was
converted into $SO_2$ in EA in the presence of $Cu_2O$. $SO_2$ from EA was ionized and $\delta^{34}S$ value was
measured using IRMS. For the determination of $\delta^{18}O$, $BaSO_4$ pyrolysis was conducted in graphite
furnace at 1450 ℃, and $\delta^{18}O$ value was obtained in CO produced from the pyrolysis at continuous-flow
mode. The results of $\delta^{34}S$ and $\delta^{18}O$ were with respect to international standard V-CDT and V-SMOW,
and the accuracy were better than ±0.2‰ and ±0.3‰, respectively.
**3 Results and discussion**
3.1 Concentrations of $PM_{2.5}$, sulfate and $SO_2$



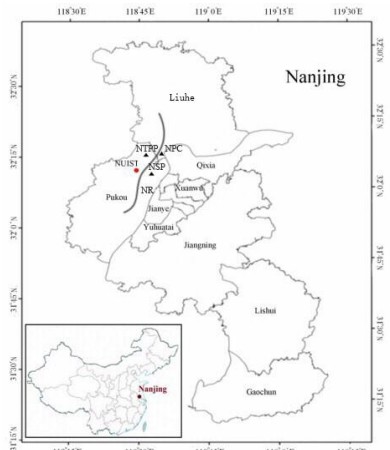


Fig.1. Sampling site of NUIST in Nanjing, China. NSP: Nanjing steel plants; NTPP: Nanjing

thermal power plants; NPC: Nanjing petrochemical company; NR: Ningliu Road.

As described in Fig. 2, the mass concentrations of $PM_{2.5}$, $SO_4^{2-}$ and $SO_2$ during the period from 4 to
22 Dec. 2019 in NUIST changed from 28.1 to 67.0 μg m$^{-3}$, 8.3 to 17.8 μg m$^{-3}$ and 6.2 to 20.9 μg m$^{-3}$
with an average and standard deviation at 45.7±12.1 μg m$^{-3}$, 12.7±3.3 μg m$^{-3}$ and 10.2±4.4 μg m$^{-3}$,
respectively. It can be observed that $PM_{2.5}$ average concentration was about 1.3 times of the First Grade
National Ambient Air Quality Standard (35 μg m$^{-3}$) and beyond the safety standard of World Health
Organization (10 μg m$^{-3}$). The photochemical reactivity during the winter in Beijing has been found to
be relatively high (Zhang et al., 2020), which facilitates the formation of some photooxidants. The
relatively clean days during the sampling period indicates the importance of photoinduced oxidation of
$SO_2$.



**Fig. 2.** The concentrations of $PM_{2.5}$, $SO_4^{2-}$ and $SO_2$.

Meanwhile, the change trends of $PM_{2.5}$, $SO_4^{2-}$ and $SO_2$ concentrations were found to be basically the

same during the sampling period, suggesting sulfate was mainly from $SO_2$ oxidation. Especially, $PM_{2.5}$,

$SO_4^{2-}$ and $SO_2$ concentrations increased to the maximum values on 10 Dec.. It is noted that $NO_2$ and

CO concentrations were 85 and 1.60 μg m$^{-3}$ on 10 Dec., which were also the maximum values during

the sampling period. High CO concentration indicates that the pollution was mainly from local

emissions. However, $O_3$ concentration on 10 Dec. was the minimum value at 24 μg m$^{-3}$, which

preliminarily indicates that $SO_2$ oxidation by $NO_2$ might be a major pathway in sulfate formation.

Previous studies showed that $SO_2$ oxidation by $NO_2$ in aerosol water dominated heterogeneous sulfate

formation during wintertime at neutral aerosol pH (Wang et al., 2016; Cheng et al., 2016). However,

subsequent studies showed that the calculated aerosol pH was in the range of 4.2~4.7, and the

reactions between $SO_2$ and $NO_2$ during this pH range were too slow to produce sulfate. Taking into

account low aerosol pH in Nanjing region, we suggested that $SO_2$ oxidation by $NO_2$ was not a

dominant pathway for sulfate formation during the sampling period.

In contrast, $PM_{2.5}$, $SO_4^{2-}$ and $SO_2$ concentrations were observed to be at the minimum values On 6

Dec.. Similarly, $NO_2$ and CO concentrations were also at the minimum of 36 and 0.6 mg m$^{-3}$,

respectively. However, $O_3$ concentration on 6 Dec. was the maximum at 50 μg m$^{-3}$. Besides, the rate of

$SO_2$ oxidation with $O_3$ becomes fast only when pH>5, the reaction rate of $SO_2$ with $O_3$ is one hundredth

of those with $H_2O_2$ or TMI when pH<5. Therefore, pH values of actual fine particles at 4~5 in Nanjing

could markedly restrain $SO_2$ oxidation by $O_3$. The lowest $SO_4^{2-}$ concentration on 6 Dec. further

demonstrated that $SO_2$ oxidation by $O_3$ played an insignificant role in sulfate formation.

Generally, aqueous-phase oxidation is deemed to be a main process of sulfate formation in

atmospheric environment. Shao et al. (2018) believed that heterogeneous sulfate production on aerosols

occurred when relative humidity (RH) was higher than 50 %. The RH values of the atmosphere ranging

from 50.7 to 88.9% during the sampling period indicated that sulfate formation was closely related to

the heterogeneous oxidation of $SO_2$.

3.2 Sulfur isotope compositions in sulfate and $SO_2$

It can be observed from Fig. 3 that the values of $\delta^{34}S$-$SO_4^{2-}$ were generally higher compared to those

of $\delta^{34}S$-$SO_2$ during the sampling period except that on 16 Dec.. The $\delta^{34}S$-$SO_4^{2-}$ values ranged from 3.1




to 4.7‰ with an average and standard deviation at 4.0±0.6‰, while $\delta^{34}S$-$SO_2$ values changed from -2.9
to 4.7‰ with an average and standard deviation at -0.2±2.3‰. The discrepancy between the values of
$\delta^{34}S$-$SO_4^{2-}$ and $\delta^{34}S$-$SO_2$ was mainly related to sulfur isotope fractionation effect during $SO_2$ oxidation
to secondary sulfate.

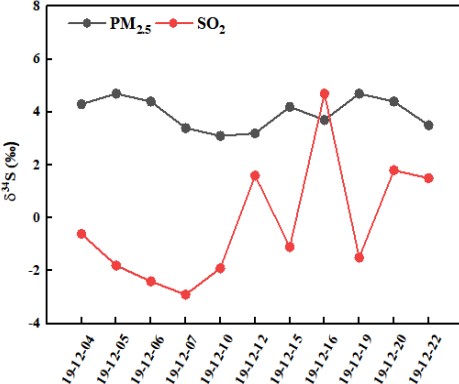


**Fig. 3.** Sulfur isotope compositions in sulfate and $SO_2$.

It is noteworthy that the values of $\delta^{34}S$-$SO_4^{2-}$ were similar to that in $PM_{2.5}$ with an average at 4.2‰

during Youth Olympic Games in Aug. 2014 in Nanjing (Guo et al., 2016). However, the average value
of $\delta^{34}S$-$SO_4^{2-}$ during the sampling period was lower than 5.6‰ in Nanjing during a typical haze event
from 21 Dec. 2015 to 1 Jan. 2016 (Guo et al., 2019). The higher $\delta^{34}S$ values of sulfate in haze was
possibly ascribed to $SO_2$ heterogeneous oxidation, which typically enriched heavy sulfur isotope in
sulfate. In this study, the average concentrations of $PM_{2.5}$ was 45.7 μg m$^{-3}$, indicating a not heavily
polluted time interval. Besides, the relatively high temperature during the sampling period was
favorable for photochemical reactions and OH radicals formation. As a result, the contribution of $SO_2$
homogenous oxidation increased during sulfate formation, which enriched light sulfur isotope
compared to that in haze. Han et al. (2017) determined $\delta^{34}S$ values in Beijing $PM_{2.5}$ with an average at
6.0‰. It is observed that there existed a regional difference in $\delta^{34}S$-$SO_4^{2-}$ values. The $\delta^{34}S$-$SO_4^{2-}$ in
Nanjing was generally lower than that in Beijing. The discrepancy of $\delta^{34}S$-$SO_4^{2-}$ illustrated different
sulfur sources and $SO_2$ oxidation pathways in these regions. In addition, $\delta^{34}S$-$SO_4^{2-}$ values presented a
seasonal change. $\delta^{34}S$ values in Beijing aerosol sulfate varied from 3.4 to 7.0‰ with an average of
5.0‰ in summer and from 7.1 to 11.3‰ with an average of 8.6‰ in winter. Generally, the
homogeneous oxidation of $SO_2$ dominated in summer compared to that in winter due to strong solar
irradiation (Han et al., 2016). $SO_2$ oxidation might lead to sulfur isotope fractionation, which was
mainly attributed to equilibrium or kinetic discrimination between $SO_2$ and sulfate. The influence of
different oxidants on sulfur isotope fractionation needed to be further investigated.
Fig.4 presents the relationship between $\delta^{34}S\text{-}SO_4^{2-}$ and atmospheric temperature during the
sampling period. It can be observed that there existed an obviously negative correlation. The higher
temperature generally corresponded to the lower $\delta^{34}S\text{-}SO_4^{2-}$. This is mainly ascribed to kinetic effect of
sulfur isotope fractionation during $SO_2$ oxidation. At high temperature, more OH radicals were
produced and the contribution of $SO_2$ homogeneous oxidation increased. It is reported that sulfur
isotope fractionation about $SO_2$ was -9‰ for homogeneous oxidation process (Tanaka et al., 1994).
Therefore, low $\delta^{34}S$ value in sulfate at high temperature was chiefly due to elevated $SO_2$ homogeneous
oxidation.

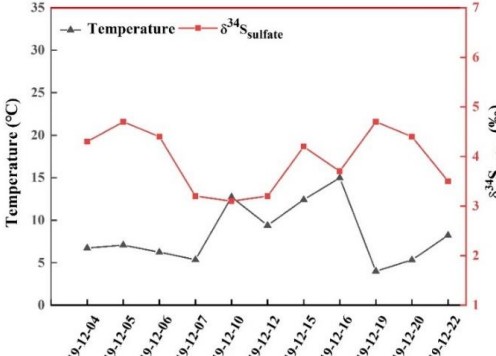


**Fig. 4.** The relationship between $\delta^{34}S\text{-}SO_4^{2-}$ and atmospheric temperature.
3.3 Sulfur isotope fractionation during $SO_2$ oxidation
The secondary sulfate was generally from $SO_2$ homogeneous and heterogeneous oxidation (Seinfeld
et al., 2016). The homogeneous and heterogeneous oxidation of $SO_2$ led to sulfur isotope fractionation,
which is described by using fractionation coefficient (α)

$$\alpha = \frac{\dfrac{\delta^{34}S_{SO_4^{2-}}}{10^3} + 1}{\dfrac{\delta^{34}S_{SO_2}}{10^3} + 1} \tag{1}$$


Sulfate enriched heavy sulfur isotope (α>1) during $SO_2$ heterogeneous oxidation due to the presence
of isotope equilibrium fractionation and kinetic fractionation. However, sulfate enriched light sulfur



isotope (α<1) during SO$_2$ homogeneous oxidation for this process was only related to kinetic
fractionation. As described in Fig. 5, α values ranged from 0.9988 to 1.0201, indicating there existed
SO$_2$ homogeneous and heterogeneous oxidation during the sampling period. α value was at the
minimum of 0.9988 on 16 Dec., which showed SO$_2$ homogeneous oxidation played a crucial role.

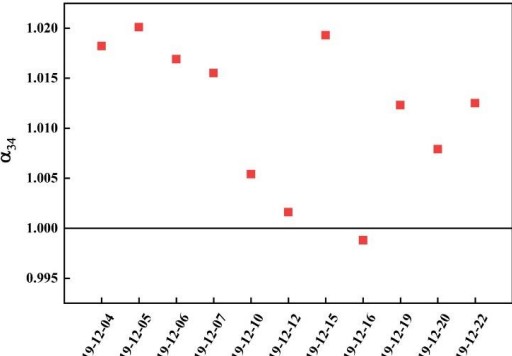


**Fig. 5.** Sulfur isotope fractionation coefficients during SO$_2$ oxidation.

It is reported that sulfur isotope fractionations during SO$_2$ heterogeneous and homogeneous oxidation

to sulfate were 16.5‰ and -9‰, respectively (Tanaka et al., 1994). Consequently, the contribution of
SO$_2$ heterogeneous and homogeneous oxidation to sulfate could be calculated by sulfur isotope mass
equilibrium equations (2) and (3).

$\delta^{34}S_{SO_2}+16.5x-9y=\delta^{34}S_{SO_4^{2-}}$                        (2)

$x+y=1$                        (3)

where x and y represent the contribution of SO$_2$ heterogeneous and homogeneous oxidation,
respectively.

It is observed from Fig. 6 that the contribution of SO$_2$ heterogeneous oxidation markedly fluctuated

ranging from 31.4 to 62.0% with an average and standard deviation at 51.6±0.1%, which indicated that
SO$_2$ heterogeneous oxidation was generally dominant during sulfate formation. He et al. (2018)
presented the observations of oxygen-17 excess of PM$_{2.5}$ sulfate collected in Beijing haze from Oct.
2014 to Jan. 2015, and found the contribution of heterogeneous sulfate production was about 41~54%
with a mean of 48±5%. The contribution of SO$_2$ heterogeneous oxidation reached high-level during 5-7
Dec. and on 19 Dec., which was closely related to the temperature of the atmosphere. The low
temperature about 5 ℃ in these days was favorable for SO$_2$ dissolution in water and further oxidized to



sulfate by the oxidants. On 16 Dec., the contribution of $SO_2$ heterogeneous oxidation was the minimum
at 31.4%. The highest temperature of 15 °C on 16 Dec. restrained $SO_2$ solubility in aqueous solution
and produced lots of gaseous oxidants such as  OH to promote $SO_2$ homogeneous oxidation.

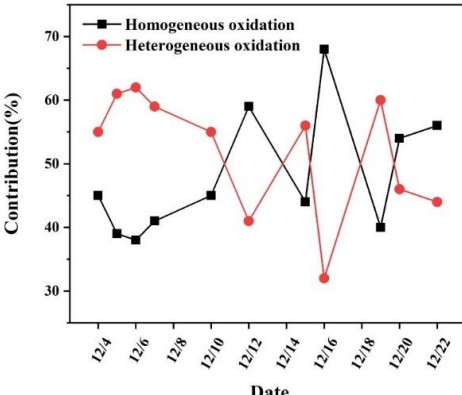


**Fig. 6.** The contributions of $SO_2$ heterogeneous and homogeneous oxidation to sulfate.

Overall, the temperature was an important factor in controlling $SO_2$ oxidation pathways. High

temperature facilitated kinetic fractionation of sulfur isotope during $SO_2$ oxidation to sulfate, thereby
decreasing $\delta^{34}S$ value in sulfate. In addition, it was not found to be positive correlation between the
contribution of $SO_2$ heterogeneous oxidation and $O_3$ or $NO_2$ concentration. This also further
demonstrated that $SO_2$ oxidation by $O_3$ and $NO_2$ were not main pathways during the sampling period.
Consequently, we mainly focused on $SO_2$ heterogeneous oxidation by $H_2O_2$ and $Fe^{3+}/O_2$ in the
following study.
3.4 The correlation of $\delta^{18}O$ between $H_2O$ and $SO_4^{2-}$ from $SO_2$ oxidation by $H_2O_2$ and $Fe^{3+}/O_2$

It is known that $SO_2$ rapidly equilibrates with ambient water for very high molar ratio of $H_2O$ to $SO_2$

in the atmosphere. As a result, $\delta^{18}O$ of $SO_2$ is dynamically controlled by $\delta^{18}O$ of water and $\delta^{18}O$ of $SO_2$
has no obvious effect on $\delta^{18}O$ of sulfate formed by different oxidation pathways. Meanwhile, sulfate is
very stable with respect to O atom exchange with ambient water. Consequently, $\delta^{18}O$ can be adopted to
distinguish $SO_2$ oxidation processes due to that $\delta^{18}O$ of product sulfate reflected the distinctive signals
of different oxidants.

In this manuscript, we firstly studied $SO_2$ heterogeneous oxidation by $H_2O_2$ and $Fe^{3+}/O_2$ in the lab,

which aims to make clear the relationship of $\delta^{18}O$ between product sulfate and water at 10 °C. It can be



observed from Fig. 7 that $\delta^{18}O$ of sulfate was linearly dependent on $\delta^{18}O$ of water, and the slope of
linear curve for $H_2O_2$ oxidation approximates a ratio of 0.43, indicating that the isotopy of about two of
four oxygen atoms in sulfate was controlled by $\delta^{18}O$ of water. The other two oxygen atoms were from
$H_2O_2$ molecules, whose O-O bonds remained intact during $SO_2$ oxidation. In addition, we noted from
Fig. 7 that the slope of linear curve for $Fe^{3+}/O_2$ oxidation was 0.65, which represented that the isotopy
of about three of four oxygen atoms in sulfate was related to $\delta^{18}O$ of water. A 3/4 control of sulfate
oxygens by water is also characteristic of heterogeneous oxidation mechanisms in which $HSO_3^-$
isotopically equilibrated with water prior to significant oxidation to $SO_4^{2-}$. The other one oxygen atom
in sulfate was from $O_2$. The higher slope suggested a higher dependence of $\delta^{18}O$ of sulfate on $\delta^{18}O$ of
water during $SO_2$ heterogeneous oxidation by $Fe^{3+}/O_2$. The difference of the slope for different $SO_2$
heterogeneous oxidation processes provides us a novel method to distinguish $SO_2$ oxidation pathways.

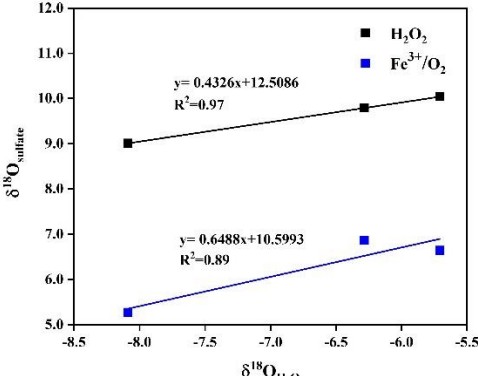


**Fig.7.** The correlation of $\delta^{18}O$ between $H_2O$ and $SO_4^{2-}$ from $SO_2$ oxidation by $H_2O_2$ and $Fe^{3+}/O_2$,
respectively.
3.5 $\delta^{18}O$-$SO_4^{2-}$ in $PM_{2.5}$ and $SO_2$ main oxidation pathways
As depicted in Fig. 8, $\delta^{18}O$ values of sulfate in $PM_{2.5}$ ranged from 11.09 to 12.93‰ with an average
and standard deviation of 12.35±0.68‰. $\delta^{18}O$ values of sulfate focused on a narrow scope except those
on 5 and 22 Dec.. It should be pointed out $\delta^{18}O$ value of secondary sulfate was a comprehensive result
from different $SO_2$ oxidation processes. Sulfate in $PM_{2.5}$ usually consisted of primary sulfate and
secondary sulfate. The $\delta^{18}O$ value of primary sulfate is about 38 ‰, which is significantly higher than
those of secondary sulfates. The contribution of primary and secondary sulfate in the atmosphere can



be calculated by oxygen isotope mass equilibrium equation (4) (Ben et al., 1982).
$$\delta^{18}O_{PM_{2.5}}=\delta^{18}O_{PS}\times(1-f_{SS}) + \delta^{18}O_{SS}\times f_{SS} \tag{4}$$
where $\delta^{18}O_{PM_{2.5}}$, $\delta^{18}O_{PS}$ and $\delta^{18}O_{SS}$ mean $\delta^{18}O$ values of PM$_{2.5}$, primary sulfate and secondary sulfate,
respectively; $f_{SS}$ is the contribution of secondary sulfate in PM$_{2.5}$.

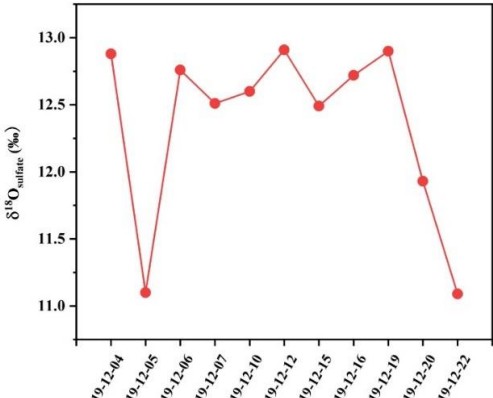


**Fig.8.** $\delta^{18}O$ values of sulfate in PM$_{2.5}$ during the sampling period.
Table 1 shows the contribution of primary sulfate and secondary sulfate in PM$_{2.5}$ during the sampling
period. It can be observed that the majority of sulfate in PM$_{2.5}$ was secondary sulfate. Secondary sulfate
appears to constitute from 80.0 to 86.1% of the total sulfate. As discussed above, secondary sulfate was
mainly ascribed to SO$_2$ homogeneous oxidation by OH radicals and heterogeneous oxidation by H$_2$O$_2$
and Fe$^{3+}$/O$_2$. Therefore, it is admirable to quantitively describe these formation pathways of secondary
sulfate in PM$_{2.5}$.
**Table 1** The contribution of primary sulfate and secondary sulfate in PM$_{2.5}$.

| Sampling time | Primary sulfate (%) | Secondary sulfate (%) |
|---|---|---|
| 4 Dec. | 10.9-23.7 | 76.3-89.1 |
| 5 Dec. | 4.6-18.2 | 81.8-95.4 |
| 6 Dec. | 10.6-23.3 | 76.7-89.4 |
| 7 Dec. | 9.6-22.5 | 77.5-90.4 |
| 10 Dec. | 10.0-22.8 | 77.2-90.0 |
| 12 Dec. | 11.1-23.8 | 76.2-89.9 |
| 15 Dec. | 9.6-22.5 | 77.5-90.4 |



| 16 Dec. | 11.9-23.6 | 76.4-88.1 |
| 19 Dec. | 11.0-23.7 | 76.3-89.0 |
| 20 Dec. | 7.7-20.8 | 79.2-92.3 |
| 22 Dec. | 4.5-18.1 | 79.1-95.5 |

It is noteworthy that there exists a linear relationship between $\delta^{18}O$ values in water and primary
sulfate or secondary sulfate from different oxidation pathways (Fig. 9), which can be described by the
equations (5)-(8). $\delta^{18}O$ values of sulfate in atmospheric samples consist of those of primary sulfate and
secondary sulfate. Considering the contribution of primary sulfate and secondary sulfate as well as $\delta^{18}O$
water is about -6.2‰ in Nanjing region, we can calculate the ratios of different $SO_2$ oxidation pathways
at 10 ℃ via oxygen isotope mass equilibrium equations (9)-(11), and the corresponding results are
depicted in Table 2.

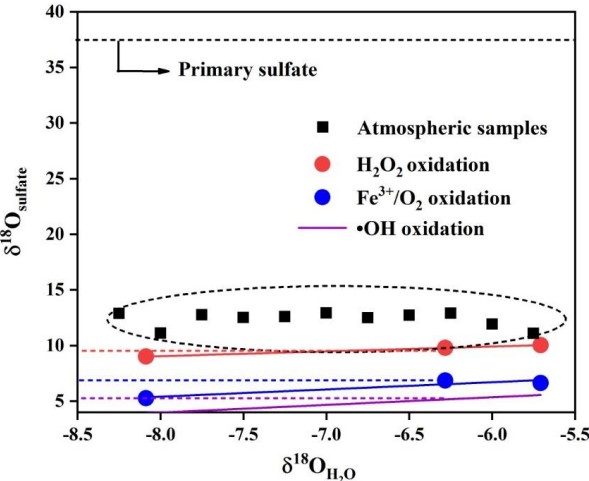


**Fig.9.** The correlation between $\delta^{18}O$ values in water and sulfate in $PM_{2.5}$.
$\delta^{18}O_{sulfate}=0.06\times\delta^{18}O_{water}+38$ ‰(PS) (Holt et al., 1983)          (5)
$\delta^{18}O_{sulfate}=0.69\times\delta^{18}O_{water}+9.5$ ‰ (SS, OH) (Holt et al., 1983)          (6)
$\delta^{18}O_{sulfate}=0.65\times\delta^{18}O_{water}+10.6$ ‰ (SS, $Fe^{3+}/O_2$) (this study)          (7)
$\delta^{18}O_{sulfate}=0.43\times\delta^{18}O_{water}+12.5$ ‰ (SS, $H_2O_2$) (this study)          (8)
$\delta^{18}O_{PM_{2.5}}=\delta^{18}O_{PS}\times f_{PS}+ (\delta^{18}O_{SS-OH}\times f_{SS-OH}+\delta^{18}O_{SS-Fe^{3+}/O_2}\times f_{SS-Fe^{3+}/O_2}+\delta^{18}O_{SS-H_2O_2}\times f_{SS-H_2O_2})\times f_{SS}$          (9)
$f_{PS}+ f_{SS}=1$          (10)



$f_{SS-OH} + f_{SS-Fe^{3+}/O_2} + f_{SS-H_2O_2} = 1$                                                                 (11)
where $\delta^{18}O_{PM_{2.5}}$ and $\delta^{18}O_{PS}$ are $\delta^{18}O$ values of total sulfate and primary sulfate in $PM_{2.5}$; $\delta^{18}O_{SS-OH}$,
$\delta^{18}O_{SS-Fe^{3+}/O_2}$ and $\delta^{18}O_{SS-H_2O_2}$ are $\delta^{18}O$ values of secondary sulfate from $SO_2$ oxidation by OH radical,
$Fe^{3+}/O_2$ and $H_2O_2$, respectively; $f_{PS}$ and $f_{SS}$ are the contribution of primary and secondary sulfate; $f_{SS-OH}$,
$f_{SS-Fe^{3+}/O_2}$ and $f_{SS-H_2O_2}$ are the contribution of secondary sulfate from $SO_2$ oxidation by OH radicals,
$Fe^{3+}/O_2$ and $H_2O_2$, respectively.
Unlike heavily polluted days with reduced solar irradiation, the photochemical reactivity could
remain high in relatively clean days during the observation period because of intense solar irradiation.
As a result, some photochemical reactive species such as OH radicals and $H_2O_2$ are deemed to be the
major oxidants for sulfate formation. It is observed from Table 2 that the ratio of $SO_2$ oxidation by OH
radicals ranged from 38 to 68% with an average and standard deviation at 48±9.7%. The ratio reached
the maximum of 68% on 16 Dec., which is mainly ascribed to the highest temperature of 15 °C during
the sampling period. The photochemical reactions are favorable for producing more OH radicals. In
contrast, the ratio of $SO_2$ oxidation by OH radicals decreased to the minimum on 6 Dec. due to the low
temperature.

**Table 2** The ratio of $SO_2$ different oxidation pathways to sulfate.

| Sampling time | OH oxidation ratio | $H_2O_2$ oxidation ratio | $Fe^{3+}/O_2$ oxidation ratio | Percentage of $H_2O_2$ oxidation in $SO_2$ heterogeneous reactions (%) |
|---|---|---|---|---|
| 4 Dec. | 0.45 | 0.27 | 0.28 | 49 |
| 5 Dec. | 0.39 | 0.24 | 0.37 | 40 |
| 6 Dec. | 0.38 | 0.24 | 0.38 | 39 |
| 7 Dec. | 0.41 | 0.25 | 0.34 | 43 |
| 10 Dec. | 0.45 | 0.27 | 0.28 | 49 |
| 12 Dec. | 0.59 | 0.30 | 0.11 | 74 |
| 15 Dec. | 0.44 | 0.26 | 0.30 | 47 |
| 16 Dec. | 0.68 | 0.26 | 0.06 | 80 |
| 19 Dec. | 0.4 | 0.25 | 0.35 | 41 |
| 20 Dec. | 0.54 | 0.31 | 0.15 | 67 |



| 22 Dec. | 0.56 | 0.32 | 0.12 | 72 |
|---------|------|------|------|-----|

$SO_2$ heterogeneous oxidation was relatively dominant during the sampling period. It is known that $SO_2$ oxidation by $H_2O_2$ and $Fe^{3+}/O_2$ are the most important pathways during the heterogeneous oxidation. From table 2, the percentage of sulfate from $SO_2$ oxidation by $H_2O_2$ in total secondary sulfate from $SO_2$ heterogeneous oxidation reactions varied from 39 to 80% with an average and standard deviation at 54.6±15.5%, indicating that $H_2O_2$ oxidation predominated during $SO_2$ heterogeneous reactions. In addition, there existed an obviously positive correlation between the ratio of $SO_2$ oxidation by $H_2O_2$ and OH radicals, which was chiefly attributed to photochemical reactions. The relatively strong solar irradiation on 16 Dec. resulted in the maximum ratio of 80% about $H_2O_2$ oxidation in $SO_2$ heterogeneous reactions. The sampling site is near to Nanjing steel plant. As companion emitters, $Fe^{3+}$ are present in much higher concentrations than that in other areas. It is believed that $SO_2$ oxidation by $O_2$ in the presence of $Fe^{3+}$ was important in the areas where the concentrations of $SO_2$ and $Fe^{3+}$ were high. This inevitably resulted in high $Fe^{3+}/O_2$ oxidation ratio in $SO_2$ heterogeneous oxidation reactions.

**4 Conclusions**

There was no serious $PM_{2.5}$ pollution during the sampling period. The secondary sulfate constitutes from about 80.0 to 86.1% of total sulfate in $PM_{2.5}$. $SO_2$ oxidation by $O_3$ and $NO_2$ played an insignificant role in sulfate formation. The secondary sulfate was mainly ascribed to $SO_2$ homogeneous oxidation by OH radicals and heterogeneous oxidation by $H_2O_2$ and $Fe^{3+}/O_2$. Compared to homogeneous oxidation, $SO_2$ heterogeneous oxidation was generally dominant during sulfate formation. The contribution of $SO_2$ heterogeneous oxidation was about 52%. $SO_2$ oxidation by $H_2O_2$ predominated in $SO_2$ heterogeneous oxidation reactions and the average ratio of which reached 55%.

**Author contribution**

Ziyan Guo analyzed the data and wrote the original draft. Keding Lu designed the methodology and administrated the project. Pengxiang Qiu and Mingyi Xu performed the data collection. Zhaobing Guo reviewed and revised the paper

**Competing interests**



The authors declare that they have no known competing interests or personal relationships that could
have appeared to influence the work reported in this paper.
**Acknowledgement**
We gratefully acknowledge the financial supports from the National Natural Science Foundation of
China (Nos. 41873016, 51908294, and 21976006), the National Science Fund for Distinguished Young
Scholars (No. 22325601).




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
