# Peer review of "Quantifying SO2 oxidation pathways to atmospheric sulfate by using"

_EGUsphere, 2023_

## Referee Comment (RC1)

General Comments:

In recent years, air pollution is seriously threatening the health of millions of people in China. Sulfate is one of the major chemical species in $PM_{2.5}$, and play a critical role in human health, and environmental chemistry. However, its formation in the atmosphere remains controversial. In this study, both observational data ($\delta^{34}S$ and $\delta^{18}O$ values) and laboratory simulation are used to constrain $SO_2$ oxidation pathways. The authors found that the sulfate in $PM_{2.5}$ was mainly formed from the oxidation of $SO_2$ by OH, $H_2O_2$ and TMI. This work provides a valuable dataset of $\delta^{34}S$ and $\delta^{18}O$ that add critical constraints for sulfate formation pathways.

Specific Comments:

1. Page 4, Lines 83-84 The authors mention that the sulfur isotopic fractionation factor of $SO_2$ oxidation by OH determined with laboratory experiments by Harris et al. (2012) was 1.0087. However, they discussed that "It is reported that sulfur isotope fractionation about $SO_2$ was -9‰ for homogeneous oxidation process (Tanaka et al., 1994)". The sulfur isotopic fractionation factor for homogeneous pathway ($SO_2$+OH) obtained by Tanaka et al. (1994) is different from the laboratory results by Harris et al. (2012). The authors need to compare these two values and explain which to be used for their discussion.

2. Page 17, lines 345-348 Their calculations displayed that the $H_2O_2$ pathway is predominated during heterogeneous oxidation of $SO_2$. Could the authors discuss the sources of $H_2O_2$ in atmosphere if it plays an important role in heterogeneous oxidation of $SO_2$?

3. The conclusion seems to be a bit dry. I suggest that the important implications for this work should be added, besides summarize the main points.

Technical corrections:

Page 8, Line 163 Please change "The concentrations of $PM_{2.5}$, $SO_4^{2-}$ and $SO_2$" to "Variations in concentrations of $PM_{2.5}$, $SO_4^{2-}$ and $SO_2$".

Page 9, Line 197 Please change "Sulfur isotope compositions in sulfate and $SO_2$" to "Variations in sulfur isotope compositions in sulfate and $SO_2$". In addition, the black solid circles represent the $\delta^{34}S$ values of sulfate instead of $PM_{2.5}$.

---

## Referee Comment (RC2)

This manuscript quantifies the sulfate formation pathways from 4 to 22 December 2019 in Nanjing by proposing a new method of simultaneously measuring sulfur and oxygen isotope compositions. The authors conclude that sulfate in $PM_{2.5}$ is mainly from a secondary source with $SO_2$ homogeneously oxidized by OH and heterogeneously oxidized by $H_2O_2$. Overall, the manuscript is well-written, and the method is reasonable. I have a few points that could be addressed to strengthen the manuscript and some minor comments.

General Comments:

1. The method seems applicable, but the authors need to explain the calculations better. I find it hard sometimes to understand how the result is derived. For example, the authors mentioned that the $\delta^{18}O$ value of primary sulfate is about 38 ‰ in Line 296 before they pointed out it was based on Formula (5). That is confusing. Why there are contribution ranges on each day in Table 1, instead of a single number like in Table 2?

2. Is it possible to add more data points in Figure 7? It seems three are not robust enough to rerive the linear relationships.

Minor Comments:

Line 69: Define RH here instead of in Line 186.

Line 98/320: I did not find references related to Holt et al.

Line 168-170: I am not sure why high CO is indicative of local emissions. It can be transported by a long range.

Line 192: What does the negative -2.9 mean here? Is it possible to have negative values?

Figure 3: Legend of PM2.5 is wrong. Should be sulfate.

Line 249: The average of 51.6% seems just a little higher than 50%. I suggest to say that most of the days (seems 7 out of 11) have more than 50% contributions from heterogeneous oxidation.

Figure 7: Are the three dots corresponding to three kinds of water? Better to describe it in the texts or figure title.

Line 324: $f_{SS-OH+}$ should be $f_{SS-OH}+$

---

## Author Response (AR1)

<h1 style="text-align:center">Response to Reviewer#1's comments</h1>

**General Comments:**

In recent years, air pollution is seriously threatening the health of millions of people in China. Sulfate is one of the major chemical species in $PM_{2.5}$, and play a critical role in human health, and environmental chemistry. However, its formation in the atmosphere remains controversial. In this study, both observational data ($\delta^{34}S$ and $\delta^{18}O$ values) and laboratory simulation are used to constrain $SO_2$ oxidation pathways. The authors found that the sulfate in $PM_{2.5}$ was mainly formed from the oxidation of $SO_2$ by OH, $H_2O_2$ and TMI. This work provides a valuable dataset of $\delta^{34}S$ and $\delta^{18}O$ that add critical constraints for sulfate formation pathways.

**Specific Comments:**

1. Page 4, Lines 83-84 The authors mention that the sulfur isotopic fractionation factor of $SO_2$ oxidation by OH determined with laboratory experiments by Harris et al. (2012) was 1.0087. However, they discussed that "It is reported that sulfur isotope fractionation about $SO_2$ was -9‰ for homogeneous oxidation process (Tanaka et al., 1994)". The sulfur isotopic fractionation factor for homogeneous pathway ($SO_2$+OH) obtained by Tanaka et al. (1994) is different from the laboratory results by Harris et al. (2012). The authors need to compare these two values and explain which to be used for their discussion.

**Response:** Thanks for Reviewer's rigorous work. Harris et al. (2012) measured sulfur isotopic fractionation factor ($\alpha OH$) of $SO_2$ oxidation by OH radicals, which was from the photolysis of water vapor at 30% relative humidity and 184.9 nm. They found that $\alpha OH$ was negatively correlated to the temperature and described as $\alpha OH=(1.0089\pm0.0007)-((4\pm5)\times10^{-5})T$ (°C). In the revised manuscript, we cancelled $\alpha OH$ values of $SO_2$ oxidation by OH, $O_3/H_2O_2$ and iron catalysis and emphasized their differences of $\alpha OH$ values.

In contrast, Tanaka et al. (1994) estimated $\alpha hom$ to be 0.991 during homogeneous oxidation of $SO_2$ by OH radicals by Ab initio calculations using transition state theory. The discrepancy between these two values may be explained by different research methods and/or temperature-dependence of fractionation factor. Generally, sulfate enriched light sulfur isotope ($\alpha hom<1$) during $SO_2$ homogeneous oxidation for this process was only related to kinetic fractionation. Therefore, we used $\alpha hom=0.991$ and $\alpha het=1.0165$ to study the contribution of $SO_2$ heterogeneous and homogeneous oxidation to sulfate in our study.

2. Page 17, lines 345-348 Their calculations displayed that the $H_2O_2$ pathway is predominated during heterogeneous oxidation of $SO_2$. Could the authors discuss the sources of $H_2O_2$ in atmosphere if it plays an important role in heterogeneous oxidation of $SO_2$?

**Response:** $H_2O_2$ production in the relatively clean atmosphere is ascribed to self-reaction of $HO_2$ radicals that mainly come from the reactions of OH with CO and volatile organic compounds. It is favorable for $H_2O_2$ formation under the conditions of high $O_3$ concentration, strong solar irradiation, and high temperature. We have added the sources of $H_2O_2$ in the revised manuscript.

3. The conclusion seems to be a bit dry. I suggest that the important implications for this work should be added, besides summarize the main points.

**Response:** This is a constructive suggestion, and we have added the following descriptions in Conclusions in the revised manuscript "Sulfur and oxygen isotopes can be used to gain an insight into sulfate formation. Sulfur isotope compositions in $SO_2$ and sulfate were simultaneously measured to quantify the contributions of $SO_2$ homogeneous and heterogeneous oxidation. Combining field observations of oxygen isotope in the atmosphere with the linear relationships of $\delta^{18}O$ values between $H_2O$ and sulfate from different $SO_2$ oxidation processes can obtain an increased understanding of specific sulfate formation pathways. This study is favorable for deeply investigating sulfur cycle in the atmosphere".

**Technical corrections:**

4. Page 8, Line 163 Please change "The concentrations of $PM_{2.5}$, $SO_4^{2-}$ and $SO_2$" to "Variations in concentrations of $PM_{2.5}$, $SO_4^{2-}$ and $SO_2$".

**Response:** Thanks for Reviewer's suggestion. The sentence has been revised in the manuscript.

5. Page 9, Line 197 Please change "Sulfur isotope compositions in sulfate and $SO_2$" to "Variations in sulfur isotope compositions in sulfate and $SO_2$". In addition, the black solid circles represent the $\delta^{34}S$ values of sulfate instead of $PM_{2.5}$.

**Response:** Thanks for Reviewer's suggestion. The sentence has been revised in the manuscript. The black solid circles represent the $\delta^{34}S$ values of sulfate in $PM_{2.5}$, we have revised it in Fig.3.

**Response to Reviewer#2's comments**

This manuscript quantifies the sulfate formation pathways from 4 to 22 December 2019 in Nanjing by proposing a new method of simultaneously measuring sulfur and oxygen isotope compositions. The authors conclude that sulfate in $PM_{2.5}$ is mainly from a secondary source with $SO_2$ homogeneously oxidized by OH and heterogeneously oxidized by $H_2O_2$. Overall, the manuscript is well-written, and the method is reasonable. I have a few points that could be addressed to strengthen the manuscript and some minor comments.

**General Comments:**

1. The method seems applicable, but the authors need to explain the calculations better. I find it hard sometimes to understand how the result is derived. For example, the authors mentioned that the $\delta^{18}O$ value of primary sulfate is about 38 ‰ in Line 296 before they pointed out it was based on Formula (5). That is confusing. Why there are contribution ranges on each day in Table 1, instead of a single number like in Table 2?

**Response:** We are grateful for Reviewer's suggestions. We mentioned that $\delta^{18}O$ value of primary sulfate was about 38‰, which aimed to calculate the contribution of primary and secondary sulfate in the atmosphere. The $\delta^{18}O$ value of 38‰ was cited from the study of Holt and Kumar (1984), and it was not directly from Formula (5). We have added this reference in the revised manuscript.

In addition, we have further explained the calculation method about the contribution of primary and secondary sulfate in $PM_{2.5}$ and the ratios of different $SO_2$ oxidation pathways in the revised manuscript. We calculated the contribution of primary and secondary sulfate according to the equation: $\delta^{18}O_{PM_{2.5}}=\delta^{18}O_{PS}\times(1-f_{SS})+\delta^{18}O_{SS}\times f_{SS}$. It is known that secondary sulfate was mainly ascribed to $SO_2$ homogeneous oxidation by OH radicals and heterogeneous oxidation by $H_2O_2$ and $Fe^{3+}/O_2$ in this study. Therefore, the values of $\delta^{18}O_{SS}$ can be obtained based on the following three equations, respectively.

$\delta^{18}O_{SS}=0.69\times\delta^{18}O_{water}+9.5$ ‰ (OH)

$\delta^{18}O_{SS}=0.65\times\delta^{18}O_{water}+10.6$ ‰ ($Fe^{3+}/O_2$)

$\delta^{18}O_{SS}=0.43\times\delta^{18}O_{water}+12.5$ ‰ ($H_2O_2$)

As a result, data ranges about the contribution of primary and secondary sulfate in $PM_{2.5}$ are presented

in the original manuscript. To keep consistent with the single ratios of $SO_2$ different oxidation pathways to sulfate in Table 2, we have calculated the average contribution of primary and secondary sulfate on each day in Table 1 in the revised manuscript.

2. Is it possible to add more data points in Figure 7? It seems three are not robust enough to rerive the linear relationships.

**Response:** When simulatively studying the linear relationship of $\delta^{18}O$ values between $H_2O$ and sulfate from $SO_2$ oxidation by $H_2O_2$ and $Fe^{3+}/O_2$ in the lab, we selected three kinds of representative water including tap-water, lake water and rainwater. The results showed that the slopes of these two linear curves were 0.43 and 0.65, respectively, which can basically reflect the characteristics of $SO_2$ heterogeneous oxidation mechanisms by $H_2O_2$ and $Fe^{3+}/O_2$.

We fully agree with the reviewer, and will provide more data points to precisely study the correlation in the following experimental design.

**Minor Comments:**

3. Line 69: Define RH here instead of in Line 186.

**Response:** According to Reviewer's suggestions, we have defined RH as "relative humidity" in Line 69 and deleted "relative humidity" in Line 187 in the revised manuscript.

4. Line 98/320: I did not find references related to Holt et al.

**Response:** We are very sorry for our negligence. The reference has been added in the revised manuscript.
Holt, B.D. and Kumar R.: Oxygen-18 study of high-temperature air oxidation of $SO_2$, Atmos. Environ., 18, 2089-2094, https://doi.org/10.1016/0004-6981(84)90194-X, 1984.

5. Line 168-170: I am not sure why high CO is indicative of local emissions. It can be transported by a long range.

**Response:** As Reviewer said, CO can be transported by a long range due to its stability, and CO is not indicative of local emissions. In the manuscript, the conclusion "High CO concentration indicates that the pollution was mainly from local emissions" was mainly ascribed to the analysis of meteorological conditions. During the sampling period, the wind speed was lower than 3m/s and there was presence of

static weather. Therefore, it is hard for CO to transport a long range. We did not explain clearly in the original manuscript, and we have added the analysis of meteorological conditions in the revised one. The detailed description was as "Based on the wind speed was lower than 3m/s and there was presence of static weather during the sampling period, we believed that high CO concentration was mainly from local emissions.".

6. Line 192: What does the negative -2.9 mean here? Is it possible to have negative values?

**Response:** Thanks for Reviewer's suggestion. The negative -2.9 means that the lighter sulfur isotopes were enriched in $SO_2$. It is common to have negative $\delta^{34}S$ values in the samples.

7. Figure 3: Legend of $PM_{2.5}$ is wrong. Should be sulfate.

**Response:** We are very sorry for our cursoriness. The legend in Figure has been revised.

8. Line 249: The average of 51.6% seems just a little higher than 50%. I suggest to say that most of the days (seems 7 out of 11) have more than 50% contributions from heterogeneous oxidation.

**Response:** Thanks for Reviewer's suggestion. The sentence in the manuscript has been revised.

Page 11, Line 249-251: It is observed from Fig. 6 that most of the days (7 out of 11) have more than 50% contributions from $SO_2$ heterogeneous oxidation, which indicated that $SO_2$ heterogeneous oxidation was generally dominant during sulfate formation.

9. Figure 7: Are the three dots corresponding to three kinds of water? Better to describe it in the texts or figure title.

**Response:** Thanks for Reviewer's suggestion. Three dots are corresponding to three kinds of water in Fig.7, and we have made a detailed explanation as "which aims to make clear the relationship of $\delta^{18}O$ values between product sulfate and three kinds of water at 10 °C" in Line 278 in the text.

10. Line 324: $f_{SS\text{-}OH+}$ should be $f_{SS\text{-}OH}+$

**Response:** Thanks for Reviewer's suggestion. The formula has been revised in the manuscript.